# Impact of Socioeconomic Deprivation on the Local Spread of COVID-19 Cases Mediated by the Effect of Seasons and Restrictive Public Health Measures: A Retrospective Observational Study in Apulia Region, Italy

**DOI:** 10.3390/ijerph191811410

**Published:** 2022-09-10

**Authors:** Nicola Bartolomeo, Massimo Giotta, Silvio Tafuri, Paolo Trerotoli

**Affiliations:** Interdisciplinary Department of Medicine, University of Bari Aldo Moro, 70121 Bari, Italy

**Keywords:** COVID-19, social inequalities, deprivation index, incidence, restrictive public health measures, local spread, generalized estimating equation model

## Abstract

The aim of this study was to investigate the spatiotemporal association between socioeconomic deprivation and the incidence of COVID-19 and how this association changes through the seasons due to the existence of restrictive public health measures. A retrospective observational study was conducted among COVID-19 cases that occurred in the Apulia region from 29 February 2020 to 31 December 2021, dividing the period into four phases with different levels of restrictions. A generalized estimating equation (GEE) model was applied to test the independent effect of deprivation on the incidence of COVID-19, taking into account age, sex, and regional incidence as possible confounding effects and covariates, such as season and levels of restrictions, as possible modifying effects. The highest incidence was in areas with a very high deprivation index (DI) in winter. During total lockdown, no rate ratio between areas with different levels of DI was significant, while during soft lockdown, areas with very high DI were more at risk than all other areas. The effects of social inequalities on the incidence of COVID-19 changed in association with the seasons and restrictions on public health. Disadvantaged areas showed a higher incidence of COVID-19 in the cold seasons and in the phases of soft lockdown.

## 1. Introduction

Since the emergence of SARS-CoV-2, as of 31st December 2021, there had been 6.1 million cases of infection and 1300 deaths registered in Italy [1]. Moreover, since February 2020, Italy has been one of the countries with the highest diffusion of SARS-CoV-2 cases [2,3]. The COVID-19 crisis has emphasized the inequalities between countries [4]. Understanding the impact of inequalities is important to improve strategies that mitigate the consequences of the pandemic in different countries around the word. A report by the Centers for Disease Control and Prevention (CDC) in the United States highlighted that the odds of SARS-CoV-2 infection in areas of high deprivation in Utah were three times greater than those of low deprivation [5]. Therefore, some factors may have an impact on the incidence of SARS-CoV-2 infection, such as race, ethnic composition, and social determinants [6].

Some studies have analyzed the association between the risk of diffusion of SARS-CoV-2 and socioeconomic status [7,8,9]. Differences in the COVID-19 mortality rate were found among nine different areas of a northern region of Italy, Emilia Romagna, during the first outbreak peak. Therefore, we can hypothesize that these differences are caused by the different socioeconomic situations of the individual areas; in fact, people who live in the most disadvantaged areas experienced the highest absolute and relative risk of dying [7]. A cross-sectional ecological study conducted in Japan analyzed the association between the outcomes of COVID-19 patients and socioeconomic characteristics, such as living conditions, employment, and educational attainment. Interesting results determined that social disparities have some implications for the outcomes of Japanese patients in the same way as in Western countries [8]. Likewise, a strong relationship was found between deprivation and COVID-19 prevalence in an American study involving metropolitan and rural areas, with rural areas showing a stronger direct relationship with metropolitan areas [10]. It has often observed that there are increased numbers of deaths, along with infection with SARS-CoV-2, in areas of higher deprivation [11].

Although it is known that social inequality can have a negative impact on health outcomes, the mechanism of how this works is not yet clear [4]. To analyze the socioeconomic effect, we used a deprivation index designed by Caranci et al. [12], which evaluates the population’s level of education, unemployment, home ownership, the prevalence of single-parent families, and population density.

Variants of the virus over time have been shown to be highly contagious, allowing for wide diffusion among the population. These characteristics, together with changes in behavior and social habits through the seasons, might have modified the relationship between deprivation and COVID-19 incidence in some geographical areas. In particular, seasonality plays an important role in the spread of COVID-19 [13].

Furthermore, several studies have highlighted the influence of restrictions imposed during the pandemic on many health outcomes [14,15,16]. During the pandemic, Italy went through periods with different levels of restrictions that could have changed the influence of deprivation on the spread of the virus.

We wondered if socioeconomic inequalities played a role in the spread of the virus in the Apulia region of Italy, and if the health policies adopted during the pandemic also had an effect in this role. Thus, the aim of this study was to evaluate the spatiotemporal association between the rate of incidence of SARS-CoV-2 infection and the level of deprivation in the municipality of Apulia and how this association changes through the seasons, through periods of higher or lower circulation of the virus, and the various different restrictive public health measures adopted during the pandemic period.

We conducted a retrospective observational study among COVID-19 cases that occurred in the Apulia region, Italy from 29 February 2020 to 31 December 2021.

## 2. Materials and Methods

The study was conducted in Apulia, a region located in the southeast of Italy, with a population of around 3,926,931 (2020) people, including 1.6 million families. It is the seventh region for extension (19 k km^2^), with a population density of 200 people per km^2^, and the first for coastal length (865 km) on the mainland (Figure 1). In the last two decades, the Apulian economy has expanded more than that of the rest of southern Italy, but in 2020 the per capita gross domestic product was 36.3% lower than the national average.

### 2.1. Materials

Data on documented cases of SARS-CoV-2 infection from 29 February 2020 to 31 December 2021 were extracted from the surveillance platform IRIS (Infections Regional Information System) COVID-19 developed with the WHO Go.Data outbreak investigation tool [17] and set up to manage the pandemic emergency in Apulia. The collected information included sex, age, residence location, and date of COVID-19 test results (positive or negative). Data on the resident population were collected from the Demo Istat [18], stratified by age, class, and sex. The incidence of COVID-19 per 100,000 inhabitants (IRc) was determined as the ratio between new cases and the population multiplied by 100,000.

### 2.2. Methods

The level of socioeconomic deprivation by geographic aggregate was considered through the deprivation index. The most recent deprivation index refers to 2011 and an updated version is available from Rosano et al. [19]. For the Italian national territory as a whole, the researchers produced a deprivation index (DI) at the census level, based on 2011 census data, in the same way as the Italian index is based on the 2001 census, revising the formulation of some indicators. The DI measures the level of relative social disadvantage through a combination of characteristics of the resident population, obtained in correspondence with the population and housing census. Various features were chosen to represent the conditions of disadvantage using the following representative dimensions of deprivation: poor education, job shortages, poor housing, and family conditions.

The five indicators (X1–X5) were calculated as follows:

X_1_ = (population with an education equal to elementary school, literate or illiterate)/(population aged 6 and over) × 100;

X_2_ = (unemployed or seeking their first job)/(workforce) × 100;

X_3_ = (homes occupied by renters)/(homes occupied by resident persons) × 100;

X_4_ = (total population)/(surface (m^2^) of dwellings occupied by resident persons) × 100;

X_5_ = (single father or mother with children, in single nuclear families, with and without isolated members)/(total families) × 100.

The deprivation index is: $\mathrm{DI}=\overset{5}{\underset{i=1}{\sum }}z_{i}$ with $z_{i}=\frac{x_{i}-m_{x_{i}}}{S_{x_{i}}}$, where $m_{x}$ and $S_{x}$ are the average and the standard deviation of the indicator X.

The DI can be used to describe the social characteristics of the life context, but its main use is as a proxy for the level of individual social disadvantage, especially in areas where data at the individual level are difficult to access or unavailable. Under this assumption, we calculated the municipal DI as a weighted average of the DIs of the census sections, using the resident populations in each census section as weights.

#### 2.2.1. Graphic Method

To display the association between DI and the IRc, we designed adapted heat maps, visualizing the first variable in the columns and the second in the rows, in order. As an intuitive data-visualization technology, heat maps use color to convey the relationships among data values, facilitating the visual identification of significant association clusters. We then aggregated the municipalities of the Apulia region into twenty homogeneous deprivation areas by percentiles of the DI distribution (min to max, by 5th percentile): from class 1, low deprivation (<5th percentile), to class 20, high deprivation (>95th percentile) (Appendix A). The IRc was instead classified into eight classes by percentiles (<5th percentile, 5th–10th, 10th–25th, 25th–50th, 50th–75th, 75th–90th, 90th–95th, and >95th). Two heat maps were drawn, the first using the percentiles of the IRc distribution of the entire study period, and the second using the percentiles of the monthly IRc distributions (Appendix A) to highlight the relationships even in periods with the lowest incidence.

To further investigate the potential low-risk and high-risk geographic areas in the region, bivariate choropleth maps were created. We therefore chose to show the geographical maps referring to the months of December 2020 and September 2021 when the relationship between DI and IRc appeared stronger, with positive and negative signs, respectively.

#### 2.2.2. Statistical Method

The association between the weekly incidence of new COVID-19 cases and the level of socioeconomic deprivation was tested. The Mardia test was used to verify multivariate normality between municipal IRc and DI, and consequently the nonparametric Spearman’s correlation coefficient was calculated. Fisher’s z transformation was applied to the Spearman coefficient (r_s_), and the more accurate approximation to the distribution of z (r_s_) proposed by Bonett and Wright [20] was used to incorporate theoretical improvements in estimating the confidence interval for Spearman’s r_s_.

The possible effect of the state of socioeconomic deprivation on incidence could be mediated by the season and by the level of restrictive public health measures adopted during the 2020 and 2021 pandemic periods. Therefore, the period under examination was divided into four phases, with decreasing levels of restriction:Phase 1, from 1 March 2020 to 30 April 2020, “total lockdown”, with a high level of restrictions: ban on leaving the house except out of necessity, suspension of educational services, closure of all commercial activities and public offices [21];Phase 2, from 1 May 2020 to 15 June 2020, from 1 October 2020 to 31 December 2020, and from 15 March 2021 to 25 April 2021, “soft lockdown,” with a moderate level of restriction: ban on leaving one’s hometown except for work, suspension of educational services, closure of some commercial activities [22,23];Phase 3, from 16 June 2020 to 30 September 2020 and from 26 April 2021 to 18 May 2021, a moderate level of restriction: suspension of indoor educational services, reduction in the number of people accessing commercial activities [24];Phase 4, from 8 February 2021 to 14 March 2021 and from 19 May 2021 to 31 December 2021, low level of restrictions.

A multivariate model was applied to test the independent effect of deprivation on IRc, taking into account age, sex, and regional incidence as possible confounding effects and covariates such as season, levels of restrictions, interaction between DI and season, and interaction between DI and levels of restrictions as possible modifying effects. The general spread of the virus in the region was also included in the model as a possible confounding factor of the season and phase effects. Therefore, the weekly regional incidence of new COVID-19 cases was calculated as a proxy variable of the general spread of the virus.

Compared to heat maps, the multivariate model DI was grouped into a smaller number of classes based on the quintiles of the distribution to enable more robust and more easily interpretable estimates: very high (VH), DI less than −1.51; high (H), DI between −1.51 and −1.01; medium (M), DI between −1.01 and −0.59; low (L), DI between −0.59 and −0.02; and very low (VL), DI greater than 0.02. Age was grouped into seven classes (0–5, 6–14, 15–25, 26–45, 46–65, 66–75, and 76+ years). Since these are longitudinal data related to repeatedly measured incidence, a generalized estimating equations (GEE) model using Poisson distribution with a logarithmic link function was used. The GEE Poisson estimates the same model as the standard Poisson regression, allowing for dependence within clusters, that is, municipalities. The regression coefficients were refit, correcting iteratively for the correlation. In such models, the within-subject correlation structure is treated as a nuisance parameter. In this work, the exchangeable correlation structure was used, assuming that the correlation between events remained constant through time [25].

Applying the inverse link function, we estimated the rates and 95% confidence intervals (CI) for each level of each covariate, averaged for the levels of the other covariates. We similarly estimated the rate ratios with their 95% CI, which provides all pairwise differences among the levels of covariates. All pairwise multiple comparisons were adjusted according to the Tukey correction. All statistical significance tests had two-tailed alternative hypotheses, and *p*-values less than 0.05 were considered statistically significant. Statistical analysis was performed using SAS/STAT^®^ Statistics version 9.4 (SAS Institute, Cary, NC, USA). The graphics were completed using R software version 4.1.2 (R Core Team, Vienna, Austria) [26] with the packages “ggplot2” [27] and “dplyr” [28]. Spatial maps were created using QGIS software version 3.16.11 (QGIS Development Team) [29].

## 3. Results

The first SARS-CoV-2 infection was recorded in Puglia on 1 March 2020 in a subject that travelled from Codogno (Lombardy), the epicenter of the epidemic in Italy, to Taranto. From that moment and up to 31 December 2021, 326,188 cases were recorded (92,179 in 2020 and 234,009 in 2021), of which 51.5% (167,923) were female. The median age was 43 years (interquartile range 24–59) for males, and 44 (IQR 26–59) for females. The highest incidence in 2020 was in December, while in 2021 it was in April (Table 1).

The deprivation index (DI) for the 258 municipalities of the region varies between −1.70 and 1.52, with a median value of −0.78.

The heat map in Figure 2 shows the weekly trend of the IRc measured in the geographic areas of the region that are homogeneous in terms of socioeconomic deprivation. With the IRc categorized according to the percentiles of the entire period distribution (Figure 2a), a different intensity is observed between the first wave (March–May 2020) and the two subsequent waves (November 2020–January 2021 and March–April 2021). Subsequently, periods of a complete absence of circulation of the virus no longer occur as had happened in the months of June and July 2020. A more intense shade of red, corresponding to a higher IRc, is observed in geographical areas with DI in the higher percentiles, therefore correlating with greater socioeconomic deprivation. This evidently occurs in the period November 2020–April 2021. A strong increase in incidence, without distinction between the different levels of DI, is recorded in the last week of the series (end of December 2021), which is in correspondence with the spread of the Omicron variant of SARS-CoV-2. The high IRc of December 2021 masks the relationships with the DI in periods with lower incidence. Therefore, in the second heat map (Figure 2b), the IRc was categorized using monthly distribution as a reference for determining the percentiles. In this second heat map, in addition to the period November 2020–April 2021 when a strong association between high DI and high incidences was observed, an inverse association appears in the summer months of August and September 2021 when the highest IRc was recorded in areas with lower deprivation.

To evaluate the geographic association between the level of deprivation and the incidence of infections throughout the region, we drew two bivariate choropleth maps referring to two different months. Figure 3a represents the bivariate map of the Apulia region in December 2020, when the association between high IRc and high DI was strong; the central and northwestern areas were, at the same time, the most deprived and had high incidence. While the map in Figure 3b refers to the month of September 2021 when there was an inverse association, in this case, the areas of the southeastern coasts, mainly the tourist localities, were those with the lowest level of deprivation but the highest incidence.

The weekly correlation coefficients between IRc and DI were calculated to statistically verify the associations detected graphically, and the results are shown in Figure 4. In almost the entire period, there was a statistically significant association between greater socioeconomic deprivation of a territory and higher incidence, with the highest correlation coefficients found in the third (rs 0.60, 95% CI 0.51–0.67) and fourth weeks of December 2020, and then in the second and third weeks of October 2020. In a few weeks between August and September 2021, the correlation dropped and was not significant, but still remained positive. Even in the last week of the series, in correspondence with the diffusion of the Omicron variant of SARS-CoV-2, the correlation between DI and IRc was no longer statistically significant.

A GEE model using Poisson distribution was applied to the weekly IRc using (as covariates) age class, sex, DI class, regional incidence, season, phase (levels of restrictions), interactions between DI class and season, and interaction between DI class and phase. According to the generalized score test for type III contrasts, all covariates and interactions were statistically significant (for sex, the *p*-value was 0.015; for all other variables, it was *p* < 0.001).

The estimated IRc (cases per 100,000 inhabitants ± standard error) for males was 28.83 ± 1.2 and 28.45 ± 1.16 for females (rate ratio 1.014 [1.001–1.026]). The age classes with the highest estimated IRc were 46–65 years (60.7 ± 2.7) and 26–45 years (54.5 ± 2.3), followed by 66–75 years (26.9 ± 1.1), 76+ years (25.1 ± 1.1), 6–14 years (24.6 ± 1.2), 15–25 years (23.2 ± 1.0), and 0–5 years (12.4 ± 0.8). All pairwise comparisons were statistically significant, with the exception of the RR between the 6–14 and 15–25 age-groups and between the 6–14 and 76+ age-groups (Appendix A).

The highest estimated IRc occurred in geographic areas with very high DI (47.5 ± 3.1), and the rates decreased in areas with decreasing deprivation: high DI 33.4 ± 2, medium DI 29.3 ± 2.6, low DI 22.1 ± 1.4, and very low DI 18.8 ± 2.8. The season with the highest weekly incidence was winter (58.2 ± 2.5), followed by spring (35.9 ± 1.9), autumn (31.8 ± 1.8), and summer (10.1 ± 0.6). During the total lockdown (phase 1), the number of cases was very low (6.1 ± 0.9). The highest IRc was estimated during phase 2 of the restrictions (70.1 ± 2.1). In phases 3 and 4, the IRc and its standard errors were 33.7 ± 1.9 and 46.5 ± 1.3, respectively.

In regard to deprivation, all the rate ratios between areas with very high DI, and those less deprived were statistically significant (vs. high RR 1.42, vs. medium RR 1.62, vs. low RR 2.15, vs. very low RR 2.53). In addition, RRs were significant between the area with high DI and low (RR 1.51) and very low (RR 1.78) DI. All the comparisons between seasons were significant, with the highest RRs found in comparison with the summer season. The RRs between phases were also all significant: the population was less at risk in phase 1 than in the other three phases. The RRs between phase 2 and phases 3 and 4 were 2.08 and 1.51, respectively (Figure 5).

It is interesting to observe the results of the interactions between the level of deprivation and the seasons. The highest incidence was in areas with very high DI in winter (107.2 ± 7.5) (Table 2). For autumn, the highest RRs were estimated to be as follows: 3.83 and 3.51 in comparison with very high vs. low and very low DI, respectively. Indeed, socioeconomic deprivation was decisive in autumn and winter, as significant RRs were recorded between almost all levels of deprivation. In spring, only the RRs between areas with very high deprivation and those with low and very low deprivation were significant. The effect of the DI was not significant in the summer (Figure 6a and Appendix A).

The results of the interaction between deprivation class and restriction level showed that the highest IRc was during phase 2 in areas with very high deprivation (138.4 ± 6.9) (Table 2). During phase 1 of the restrictions, no RR between areas with different levels of deprivation was significant. Meanwhile, during phase 2, soft lockdown, areas with very high DI were more at risk than all the other areas and areas with high DI than those with low or very low DI. The highest RR was during phase 3 between very high areas and very low areas (RR 3.63). During the periods with fewer restrictions (phase 4), all the RRs calculated with respect to the areas with very low DI were significant (Figure 6b and Appendix A).

Figure 7 shows the combined effect of deprivation level, season, phase, and regional incidence on municipal incidence. It is clearly seen how the level of deprivation of an area becomes decisive in the spread of new cases of COVID-19 when the general circulation of the virus is greater (i.e., when the regional incidence increases), and during the autumn and winter periods, in which the level of pandemic restrictions was in phase 2.

## 4. Discussion

In this retrospective observational study, the relationship between the incidence of new coronavirus infections and the level of socioeconomic deprivation in the municipalities of Apulia, a southern region of Italy, was analyzed. Our study adds a new perspective to the research on socioeconomic inequalities during the COVID-19 pandemic in Italy, covering a large period of time and adding effect such modifiers as seasons, restrictive public health measures, and the spread of the virus.

The main result of this study is the positive association between deprivation and the occurrence of infections when an analysis was conducted at the municipal geographic area level. Several studies on the correlations between COVID-19 incidence and socioeconomic indicators have been conducted. Many of those who use deprivation indices indicate that social inequalities are a factor in COVID-19 [30,31,32,33].

A corresponding increase in deprivation and the number of infections found in our study was observed by Urdiales et al. [34]. They found a positive association with increased occurrence and deprivation across the whole region, but when they analyzed mortality or hospital admission, the results did not confirm the association, with no effect of deprivation on the occurrence of infections.

The analysis that we have conducted allowed us to investigate the effect of seasons on the spread of COVID-19 cases, and we found that infections were higher in areas with high deprivation during autumn and winter, but not during summer. Thus, season could be seen as a modifying effect in the relationship between deprivation and diffusion of infection in municipal geographic areas. Furthermore, there is evidence of the influence of environmental conditions, in relation to seasonal cycles, on immunity and human behavior [13]. During cold periods, such as winter, people usually stay indoors, a habitat that facilitates the transmission of diseases of the upper respiratory system [35]. Furthermore, the test-and-trace method that controls diffusion-generated data has shown a high percentage of indoor infection among contacts already infected by SARS-CoV-2. [36]. Our hypothesis is that in most deprived public areas (schools, public offices) and private (workplaces, houses) closed spaces, there is no appropriate aeration; therefore, the risk of infection appears to be higher because deprivation acts as an enhancer for the well-known risks of indoor transmission. Cities affected by a high deprivation index have a higher population density and probably a lower awareness of the use of individual protection equipment in public and crowded closed places. As stated in other research, inequalities in SARS-CoV-2 infections may be due to systemic social and economic inequalities in living or working conditions [37], in which prevention strategies, such as physical distancing or improved ventilation, are more difficult to apply or have not been implemented [38,39]. These events seem to disappear in summer, because, on the contrary, occurrences increase in less deprived areas. Looking at the maps of our study, it seems that coastal towns and cities with more tourist activities were affected by a higher level of occurrence. These happened, of course, because overcrowding in summer most likely occurred in tourist towns, but the spread of cases did not reach great numbers, because in summer, people tend to meet in open spaces.

Recently, research on the impact of government policy in the UK and its consequences in the relationship between the level of deprivation of an area and monthly COVID-19 cases, has shown deprivation as a key driver of COVID-19 outcomes, and highlights the unintended negative impact of government policy [40]. The analysis in our study was performed by dividing the time periods, so-called phases, according to the different health-policy decisions regarding restrictions and social distancing. The choice to analyze the data in phases, therefore, allowed us to better highlight the effects of socioeconomic inequalities. Thus, in phases with more restrictions (total lockdown), it seems that deprivation did not have an effect on the occurrence of cases. Our results were different from other researchers, such as Urdiales, who demonstrated that deprivation was an effective factor only during lockdown, but not before and after. The differences in the results could be related to the gradation of restrictions (no meetings allowed, no one could leave their homes, all shops were closed except for food shops, shops opened but with early closing hours, etc.) applied in the areas of our study and those applied by Urdiales. Furthermore, Apulia was a region with a lower incidence with respect to the remaining Italian regions between 10 March and 18 May 2020 (called “lockdown” by Uraldies), and it was difficult to analyze the differences among geographic areas. A possible explanation could be achieved by observing phase 2 when the incidence was higher and the level of association with deprivation was also high. During this phase, different social locations had different levels of restriction imposed upon them (descending levels: cinemas, public offices, schools, sports clubs, bars, and restaurants) with low levels of control by authorities, which requires careful individual preventive behaviors. It could therefore be hypothesized that there is a relationship between high deprivation and lower attention to individual preventive measures. In addition, during phase 3 and 4 restrictions when social distancing measures were progressively phased out, determining a further increase in incidence, the effect of deprivation on incidence was observed only between areas with extreme DI values.

The analysis of the relationship between deprivation and incidence was adjusted by sex and age, but in our analysis, the average regional incidence was entered in the model as an offset and considered an approximate value of the regional virus circulation. This choice led to a stronger relationship between areas with a simultaneously higher incidence and DI. Looking at the heat maps, we can observe that the last weeks of December 2021, when the spread of the Omicron variant was dominating, and a new wave of infections surfaced, deprivation appeared to have no relevant relationship with the occurrence of infection. To further analyze the relationship between social inequalities and virus diffusion, with Omicron variants playing a part in the increasing number of cases, we need consolidated data for the year 2022 with regard to infections and deprivation.

As a limitation, in ours and other similar ecological studies, with aggregated data at the municipal level, results and considerations should not be applied at the individual level. This could determine ecological bias, supposing that the socioeconomic level could be considered equal for all resident subjects in a specific area.

A problem in our study and in those of others is the underestimation of cases and incidence. In the first phase of the pandemic (March 2020), the data flow was not standardized, and testing and isolation were not completely ready to answer the needs of the areas and population. This underestimation of the incidence could negatively affect the evaluated relationship.

The effect of vaccination could also affect the association between deprivation and incidence. The Apulia region applied the protocol of vaccination, reaching the entire population as quickly as possible, starting with first doses in February 2021 and third doses for the at-risk population in September 2021. All the vaccination hubs were active across the whole region to complete the cycle for the entire population; however, we cannot exclude the possibility of a bias related to low rates of vaccination in some areas with extreme deprivation.

The use of a validated measure of deprivation [19] and a reliable method to estimate adjusted incidence (the GEE model) could be considered a strength. Our results could be directly comparable to other Italian areas and other international studies to define the role of social inequalities in the diffusion of SARS-CoV-2 infection. In fact, our findings are consistent with previous studies in Switzerland [41], the USA [42], India [43], England [44], and France [45], which showed a positive social gradient between deprivation and the risk of testing positive for SARS-CoV-2, with the highest risk among individuals living in the most disadvantaged areas.

## 5. Conclusions

Socioeconomic deprivation had a key role in the local diffusion of SARS-CoV-2 infections and disease. Our study has shown that social inequalities change their effect in association with seasons and with restrictions related to health policies adopted during the pandemic. As a matter of fact, most deprived areas displayed a higher incidence in cold seasons, and in “soft-lockdown” phases, when preventive behavior depended on individual awareness.

## Figures and Tables

**Figure 1 ijerph-19-11410-f001:**
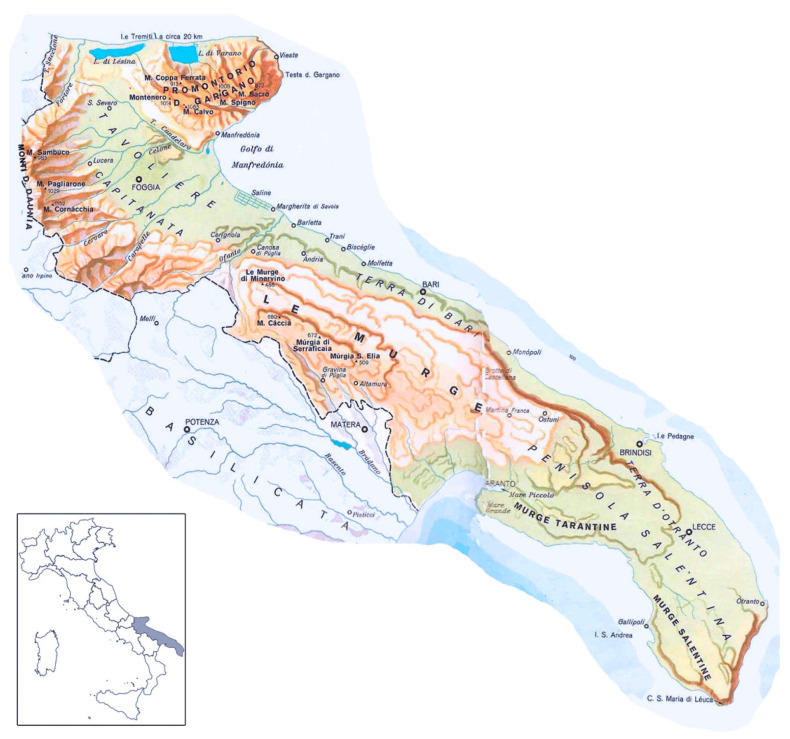
Geographic map of Apulia.

**Figure 2 ijerph-19-11410-f002:**
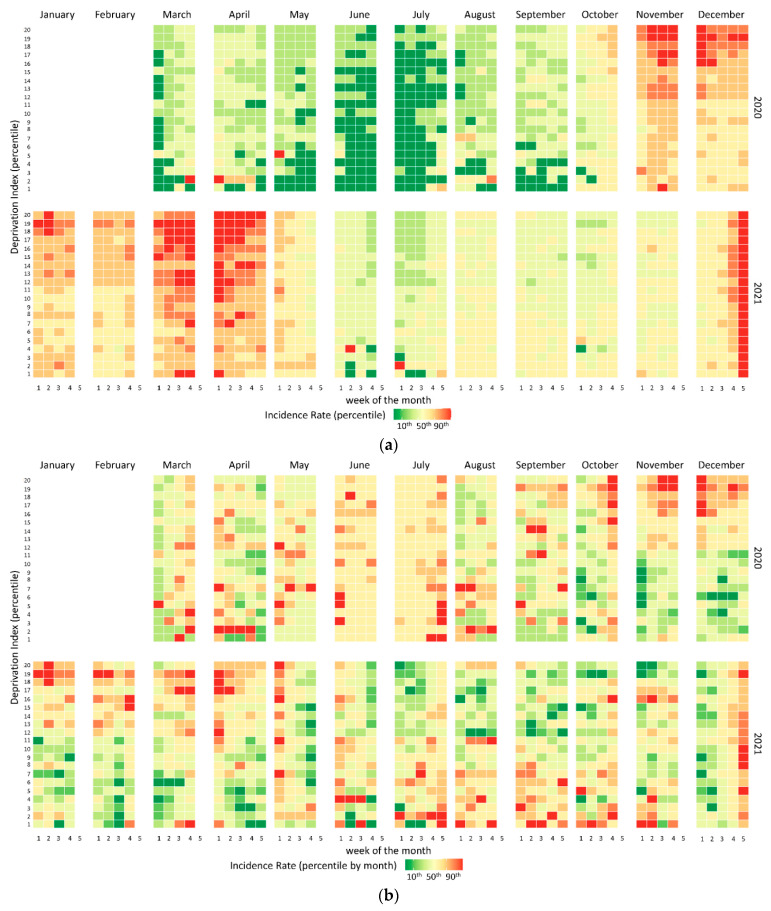
Relationships on a weekly scale between the incidence of new COVID-19 cases and the socioeconomic deprivation index, classified in percentiles. Heat maps represent their association using the percentiles of the full-period IRc distribution (**a**) and using the percentiles of the monthly IRc distribution (**b**).

**Figure 3 ijerph-19-11410-f003:**
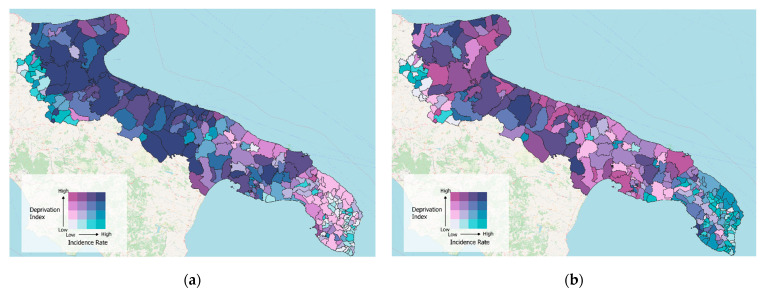
Bivariate choropleth maps of the Apulia region. Relationship between deprivation index and incidence in December 2020 (**a**) and September 2021 (**b**).

**Figure 4 ijerph-19-11410-f004:**
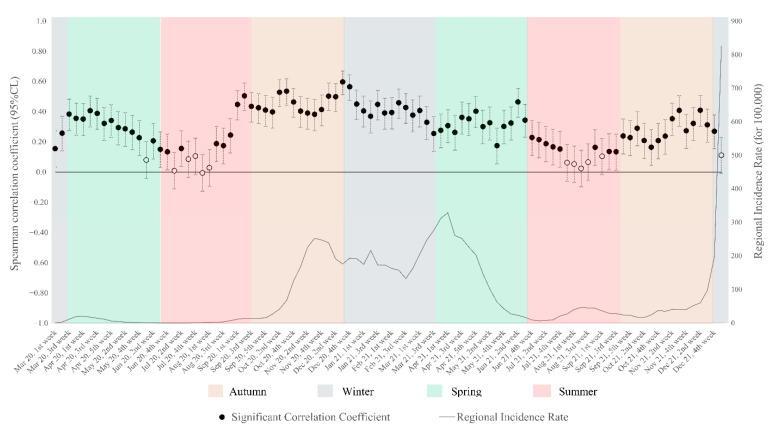
Weekly Spearman correlation coefficient between municipal incidence and deprivation index and their adjusted 95% CI, and regional incidence trend.

**Figure 5 ijerph-19-11410-f005:**
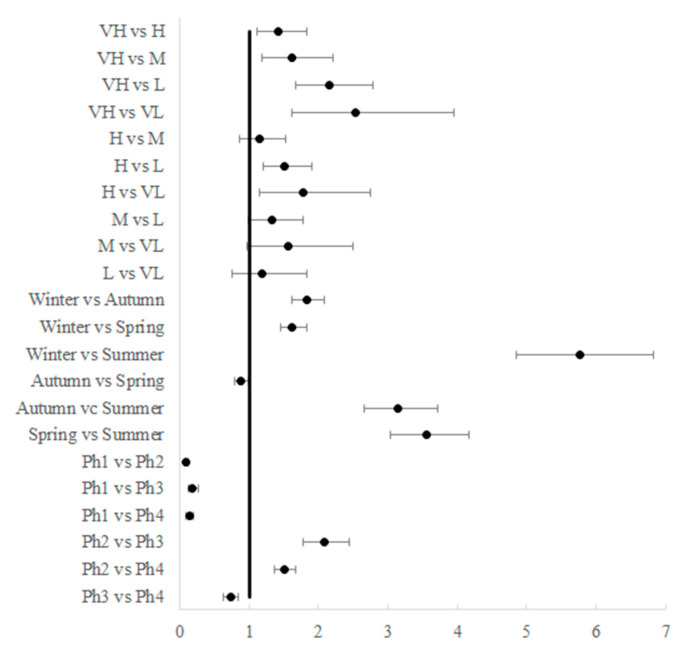
Forest plot of the rate ratios and their adjusted 95% CI between DI levels, seasons, and phases. (VH, very high DI; H, high DI; M, medium DI; L, low DI; VL, very low DI; Ph1, total lockdown; Ph2, soft lockdown; Ph3, moderate restrictions; Ph4, few restrictions).

**Figure 6 ijerph-19-11410-f006:**
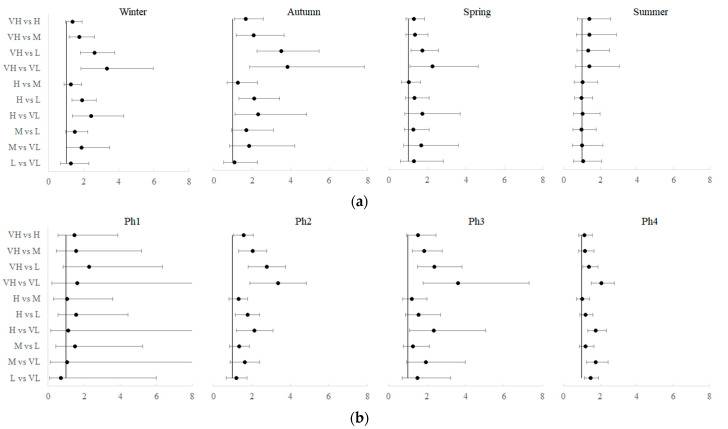
Forest plot of the rate ratios and their adjusted 95% CI between DI levels for each season (**a**) and for each phase (**b**). (VH, very high DI; H, high DI; M, medium DI; L, low DI; VL, very low DI; Ph1, total lockdown; Ph2, soft lockdown; Ph3, moderate restrictions; Ph4, few restrictions).

**Figure 7 ijerph-19-11410-f007:**
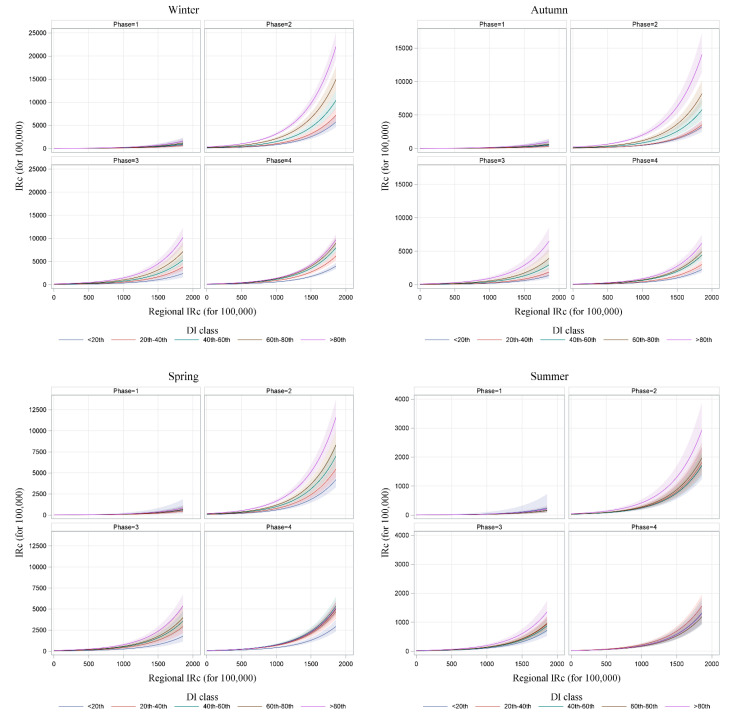
Effect plot from GEE model of the estimated local incidence by regional incidence, deprivation level, phase of the pandemic restrictions, and season.

**Table 1 ijerph-19-11410-t001:** Frequency and incidence per 100,000 inhabitants of new COVID-19 cases, by month and year.

Month	2020		2021	
	*n*	IRc	*n*	IRc
January	-	-	28,929	736.7
February	-	-	24,154	615.1
March	1334	34.0	40,870	1040.8
April	2757	70.2	49,841	1269.2
May	399	10.2	13,824	352.0
June	48	1.2	2997	76.3
July	92	2.3	2818	71.8
August	605	15.4	6897	175.6
September	2683	68.3	5504	140.2
October	10,273	261.6	3766	95.9
November	35,103	893.9	6101	155.4
December	38,877	990.0	48,286	1229.6

IRc = incidence of new COVID-19 cases per 100,000.

**Table 2 ijerph-19-11410-t002:** Estimated incidence from the GEE model for interaction effects of deprivation index by season and deprivation index by phase.

Effect		Deprivation Index				
		VH	H	M	L	VL
Season	Autumn	68.5 ± 5.3	41.2 ± 4	33 ± 4.7	19.5 ± 1.9	17.9 ± 3.3
	Winter	107.2 ± 7.5	78 ± 4.9	60.8 ± 5.3	41 ± 3.2	32.1 ± 4.8
	Spring	52.8 ± 2.9	40.8 ± 3.4	39.2 ± 3.9	30.4 ± 3	23.3 ± 4.5
	Summer	13.1 ± 1.8	9.4 ± 0.9	9.3 ± 1.3	9.8 ± 1	9.3 ± 1.5
Level of restrictions	Ph1	9.4 ± 1.7	6.4 ± 1.3	6.1 ± 1.7	4.1 ± 0.9	5.8 ± 3.3
	Ph2	138.4 ± 6.9	88.2 ± 5.3	67.5 ± 4.5	49.8 ± 3.2	41.2 ± 3.5
	Ph3	64.1 ± 4.7	41.7 ± 4.7	34.3 ± 3.1	26.8 ± 3	17.7 ± 3.3
	Ph4	61 ± 3.7	52.7 ± 3.3	52.2 ± 4	43.7 ± 2.3	29.7 ± 1.6

Data are shown as incidence of new COVID-19 cases (per 100,000) ± standard error. DI, deprivation index; VH, very high DI; H, high DI; M, medium DI; L, low DI; VL, very low DI; Ph1, total lockdown; Ph2, soft lockdown; Ph3, moderate restrictions; Ph4, few restrictions.

## Data Availability

No new data were created or analyzed in this study. Data sharing is not applicable to this article.

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
