# Peer review of "Impact of Socioeconomic Deprivation on the Local Spread of COVID-19 Cases Mediated by the Effect of Seasons and Restrictive Public Health Measures: A Retrospective Observational Study in Apulia Region, Italy"

_ijerph, 2022, doi:10.3390/ijerph191811410_

Round 1

Reviewer 1 Report

Comment 1. In page 1 (Lines 31-32), I think that a large chunk of the first paragraph of the introduction section is missing. Kindly provide a complete version of your manuscript.

Comment 2. In page (Lines 424-427). I am of the opinion that the outcome of this study should be compared with that of existing studies on other countries in Europe and other continents of the world.

Comment 3. Finally, a brief discussion of the study area (with a well annotated map) will enhance the quality of the manuscript by giving an insight to the spatial perspective of the study site.

Author Response

We thank the reviewer for his really valuable comments. We have modified the text trying to follow the reviewer's suggestions. Thanks to the suggested changes, we believe that the text has improved and we hope that this revised version of the manuscript will meet the approval of the reviewer, the editor and therefore the readers. Thank you.

Comment 1. In page 1 (Lines 31-32), I think that a large chunk of the first paragraph of the introduction section is missing. Kindly provide a complete version of your manuscript.

R. The first paragraph of the introduction was missing due to a misprint in the manuscript. We apologize to the reviewer for this error. The manuscript sent in the revised version is now complete with the missing paragraph of the introduction

Comment 2. In page (Lines 424-427). I am of the opinion that the outcome of this study should be compared with that of existing studies on other countries in Europe and other continents of the world.

R. As suggested by the reviewer, we have added a sentence at the end of the discussion that highlights the comparison between our results and studies conducted in other countries.

Comment 3. Finally, a brief discussion of the study area (with a well annotated map) will enhance the quality of the manuscript by giving an insight to the spatial perspective of the study site.

R. We thank the reviewer for the suggestion. In the materials and methods section we have added a brief description of the study area and a freeware geographic map, unfortunately all notation are in Italian language; thus, the paragraph of the discussion in which we refer to the geodemographic and economic characteristics of the region (Lines 365-370 of the first version of the manuscript. Lines 392-397 of the revised version) is better integrated.

Reviewer 2 Report

The introductory paragraph appears to be missing sentences - starts with "rate of SARS-CoV-2 infection, such..." Methods which account for strain of the virus would strengthen the research.

Author Response

We thank the reviewer for his really valuable comments. We have modified the text trying to follow the reviewer's suggestions. Thanks to the suggested changes, we believe that the text has improved and we hope that this revised version of the manuscript will meet the approval of the reviewer, the editor and therefore the readers. Thank you.

The introductory paragraph appears to be missing sentences - starts with "rate of SARS-CoV-2 infection, such..."

R. The first paragraph of the introduction was missing due to a misprint in the manuscript. We apologize to the reviewer for this error. The manuscript sent in the revised version is now complete with the missing paragraph of the introduction.

Methods which account for strain of the virus would strengthen the research.

 R. Unfortunately, the data on documented cases of SARS-CoV-2 infection were extracted from databases, which do not contain information on the virus strain. Furthermore, in the Apulia region the virus strain prevalent in the population is identified on population samples; on the basis of this information alone, for example, we have stated that since December 2021 the prevalent strain of the SARS-CoV-2 virus was “omicron”.

Reviewer 3 Report

Title:

-complete the title ,,Impact of socio-economic deprivation on the local spread of COVID19 cases mediated by the effect of seasons and restrictive public health measures. A retrospective observational study,,  with

,,Impact of socio-economic deprivation on the local spread of COVID19 cases mediated by the effect of seasons and restrictive public health measures. A retrospective observational study in Apulia Region, Italy

Abstract: we suggest to reduce this section by eliminating the lines 19-22, meaning these: The highest incidence rate was in areas with a Very High deprivation Index (DI) in 19 winter (107.2 for 100,000 ab. ± 7.5), while in autumn, the highest Rate Ratio (RR) was estimated 20 between Very High vs. Low DI (3.83, p<.001). During total lockdown, no RR between areas with 21 different levels of DI was significant, while during soft lockdown, areas with Very High DI were 22 more at risk than all other areas.

Keywords: GEE model ?? what it means ? in Section Material and methods we find that it means generalized estimating equations model but in page number 4. In Abstract section is not mentioned and maybe it will be useful and correct to mention in Abstract with its own abbreviation.

1.     Introduction

-it starts with an error because of the lack of the firsts words/sentences and with the quotation no. 6 !! (Lines 31-32)

-Lines 63-67 could be completed/adapted in a way in which include research questions, preserving the aim of the study which is clearly mentioned by the authors.

2.     Materials and Methods

- Lines 69-70 – we suggest to move these lines in Introduction section, namely completing the lines 64-65

-we suggest to insert a subtitle, named ,,2.1.Materials,, (current lines 72-78)

-insert the subtitle ,,2.2. Methods,, (current lines 79-181)

-Graphic representation - We suggest to change this subtitle with ,,Graphic method,,

-Statistical analysis - We suggest to change this subtitle with ,,Statistic method,,

3.     Results

-Fig. 1 – where are the Table S1 and Table S2 quoted inside the paper’s text?

-Fig. 3 – it is not clear where are a) and b) on this graph

4.     Discussion

-line 332, is mentioned ,,Puglia,,. Please use the same form of this toponim, namely, Apulia.

References

-the first 5 references are missing from the paper’s text

Author Response

We thank the reviewer for his really valuable comments. We have modified the text trying to follow the reviewer's suggestions. Thanks to the suggested changes, we believe that the text has improved and we hope that this revised version of the manuscript will meet the approval of the reviewer, the editor and therefore the readers. Thank you.

Title: 

-complete the title ,,Impact of socio-economic deprivation on the local spread of COVID19 cases mediated by the effect of seasons and restrictive public health measures. A retrospective observational study,,  with

,,Impact of socio-economic deprivation on the local spread of COVID19 cases mediated by the effect of seasons and restrictive public health measures. A retrospective observational study in Apulia Region, Italy

R. We thank the reviewer for the suggestion. We have changed the title as suggested.

Abstract: we suggest to reduce this section by eliminating the lines 19-22, meaning these: The highest incidence rate was in areas with a Very High deprivation Index (DI) in 19 winter (107.2 for 100,000 ab. ± 7.5), while in autumn, the highest Rate Ratio (RR) was estimated 20 between Very High vs. Low DI (3.83, p<.001). During total lockdown, no RR between areas with 21 different levels of DI was significant, while during soft lockdown, areas with Very High DI were 22 more at risk than all other areas.

R. We accept the reviewer's suggestion to reduce the abstract without completely eliminating the marked lines. We believe it is appropriate to keep a mention of the results found in order to justify the conclusions we have reached. We have therefore rephrased the sentence between lines 19-22 as follows:

The highest incidence rate was in areas with a Very High deprivation Index (DI) in winter. During total lockdown, no Rate Ratio between areas with different levels of DI was significant, while during soft lockdown, areas with Very High DI were more at risk than all other areas.

Keywords: GEE model ?? what it means ? in Section Material and methods we find that it means generalized estimating equations model but in page number 4. In Abstract section is not mentioned and maybe it will be useful and correct to mention in Abstract with its own abbreviation.

R. We have added the abbreviation "GEE" in the abstract. However, among the keywords we have reported the extended formulation "Generalized Estimating Equation model" instead of "GEE model"

1. Introduction

-it starts with an error because of the lack of the firsts words/sentences and with the quotation no. 6 !! (Lines 31-32)

R. The first paragraph of the introduction was missing due to a misprint in the manuscript. We apologize to the reviewer for this error. The manuscript sent in the revised version is now complete with the missing paragraph of the introduction.

-Lines 63-67 could be completed/adapted in a way in which include research questions, preserving the aim of the study which is clearly mentioned by the authors.

R. Accepting the referee's suggestion, we have added the following sentence to line 63 before explaining the aim of the study:

We wondered if socio-economic inequalities played a role in the spread of the virus in the Apulia region of Italy, and if the health policies adopted during the pandemic also had an effect in this role.

2. Materials and Methods

- Lines 69-70 – we suggest to move these lines in Introduction section, namely completing the lines 64-65

R. As suggested by the reviewer we have moved Lines 69-70 to the Introduction section after Line 67.

-we suggest to insert a subtitle, named ,,2.1.Materials,, (current lines 72-78)

-insert the subtitle ,,2.2. Methods,, (current lines 79-181)

-Graphic representation - We suggest to change this subtitle with ,,Graphic method,,

-Statistical analysis - We suggest to change this subtitle with ,,Statistic method,,

 R. We modified the manuscript by accepting all the referee's suggestions.

3. Results

-Fig. 1 – where are the Table S1 and Table S2 quoted inside the paper’s text?

R. Table S1 and Table S2 are cited within the text of the paper in the Method section (Graphic Method subsection) on lines 116 and 119 respectively.

-Fig. 3 – it is not clear where are a) and b) on this graph

R. We apologize to the reviewer as the title in Figure 3 was incorrect. We have entered the correct title.

4. Discussion

-line 332, is mentioned ,,Puglia,,. Please use the same form of this toponim, namely, Apulia.

R. Thanks, we corrected the name as suggested.

References

-the first 5 references are missing from the paper’s text

R. We have added the missing paragraph in the introduction; therefore, the first 5 references are no longer missing in the paper’s text.

Round 2

Reviewer 1 Report

Kindly correct this minor typo in line 35 of page 1: 'countries around the world', not word.